# No Free Lunch: Fundamental Limits of Learning Non-Hallucinating Generative Models

**Changlong Wu[1], Ananth Grama[1] & Wojciech Szpankowski[1,2]**
[1]CSoI, Purdue University   [2]Jagiellonian University
`wuchangl@hawaii.edu  {ayg,szpan}@purdue.edu`

## Abstract

Generative models have shown impressive capabilities in synthesizing high-quality outputs across various domains. However, a persistent challenge is the occurrence of "hallucinations", where the model produces outputs that are not grounded in the underlying facts. While empirical strategies have been explored to mitigate this issue, a rigorous theoretical understanding remains elusive. In this paper, we develop a theoretical framework to analyze the *learnability* of non-hallucinating generative models from a learning-theoretic perspective. Our results reveal that non-hallucinating learning is statistically *impossible* when relying solely on the training dataset, even for a hypothesis class of size two and when the entire training set is truthful. To overcome these limitations, we show that incorporating *inductive biases* aligned with the actual facts into the learning process is essential. We provide a systematic approach to achieve this by restricting the facts set to a concept class of finite VC-dimension and demonstrate its effectiveness under various learning paradigms. Although our findings are primarily conceptual, they represent a first step towards a principled approach to addressing hallucinations in learning generative models.

## 1 Introduction

Generative models have emerged as powerful tools with applications in virtually all socio-economic enterprises. At a high level, these techniques integrate large amounts of data to provide good statistical concentration and build on the resulting mixture of embeddings to produce incredible generative models of text, images, video, mechanical drawings, computer programs, mathematical proofs, and others. However, there is increasing recognition of "hallucinations" in generative models, that ultimately limit their utility in critical application settings, such as those that have correctness, accuracy, or safety constraints. Hallucinations correspond to plausible but invalid, incorrect, or misleading outputs. The key challenge in mitigating hallucinations is that there are often no characterizations of the space of "valid", "correct", or "logical" assertions, making it difficult to assess hallucinations. A common retort is that generative models should generate hypotheses that are grounded in training data. However, this assertion limits the rich potential of generative systems to that of powerful information retrieval (IR) systems, rather than those capable of new insight not grounded in training data. This tension between the desired ability of AI models to generate new hypothesis, while not generating "falsifiable" artifacts, without a well characterized notion of "true" artifacts represents the key underlying challenge of mitigating hallucinations.

Although many methods have been proposed for addressing hallucinations, such as factual data enhancement (Dziri et al., 2022), hallucination detection (Manakul et al., 2023), and fine-tuning with human feedback (Ouyang et al., 2022), their performance has primarily been validated through empirical studies, and their suitability for critical applications remains unresolved. This strongly motivates the study of hallucination and its relation to model performance from a general mathematical perspective. This paper advances the state of the art by developing a theoretical framework to understand hallucinations in generative models from a learning-theoretic perspective. We establish a rigorous theoretical foundation for analyzing the *learnability* of non-hallucinating generative models under various learning paradigms. Specifically, we characterize the conditions under which non-hallucinating learning is possible, identify the appropriate learning rules, and determine the corresponding sample complexity required to achieve learnability.

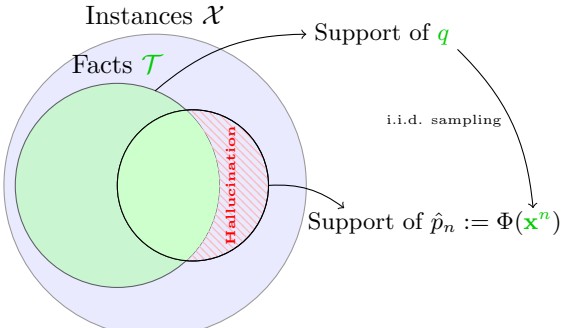

Figure 1: Illustration of the non-hallucinating learning paradigm. Here, $\mathcal{X}$ denotes the set of all sentences (factual or otherwise), and $\mathcal{T}$ denotes the set of *facts*, whose complements are *hallucinations*. The training sample $\mathbf{x}^n := \{\mathbf{x}_1, \cdots, \mathbf{x}_n\}$ is generated i.i.d. from a (faithful) *demonstrator* $q$ supported on the facts set $\mathcal{T}$. The *learned model* $\hat{p}_n$ is produced by a learning rule $\Phi$ based on $\mathbf{x}^n$, with hallucinations illustrated in *red*.

We consider the following formal setup: let $\mathcal{X}$ be an instance space; for example, in the context of language models, $\mathcal{X}$ represents the set of *all* sentences (factual or otherwise). We denote by $\mathcal{T} \subset \mathcal{X}$ a set of *facts*, which may correspond to a subset of sentences that describe true statements relevant to the task at hand. Throughout this paper, we will treat the set $\mathcal{T}$ abstractly. A generative model is represented by a distribution $p \in \Delta(\mathcal{X})$, where $\Delta(\mathcal{X})$ is the set of all probability distributions over $\mathcal{X}$ [1]. We define the *hallucination rate* of $p$ w.r.t. $\mathcal{T}$ as:

$$\mathsf{hall}(p, \mathcal{T}) = \Pr_{\mathbf{x} \sim p}[\mathbf{x} \notin \mathcal{T}]. \tag{1}$$

That is, hallucination rate measures how much probability mass a generative model assigns to *non-fact* instances. A data generation mechanism is modeled as a distribution $q \in \Delta(\mathcal{X})$, which we also refer to as a *demonstrator*. We say the demonstrator $q$ is *faithful* w.r.t. the facts set $\mathcal{T}$ if $\mathsf{hall}(q, \mathcal{T}) = 0$, i.e., the demonstrator *does not* produce non-fact samples. A learning rule is a function $\Phi : \mathcal{X}^* \to \Delta(\mathcal{X})$, where $\mathcal{X}^*$ denotes the set of all finite sequences of elements in $\mathcal{X}$. For any given $\epsilon, \delta > 0$, we say the learner $\Phi$ *non-hallucinatingly learns* $(q, \mathcal{T})$ with sample complexity $n$ at hallucination rate $\epsilon$ and confidence $\delta$, if:

$$\Pr_{\mathbf{x}^n \sim q}[\mathsf{hall}(\Phi(\mathbf{x}^n), \mathcal{T}) \geq \epsilon] \leq \delta, \tag{2}$$

where the *training* set $\mathbf{x}^n := \{\mathbf{x}_1, \cdots, \mathbf{x}_n\}$ is sampled *i.i.d.* from the demonstrator $q$.

It is crucial to note that the learner $\Phi$, in general, *does not* know the tuple $(q, \mathcal{T})$, and the only available information comes from the training set $\mathbf{x}^n$. However, the construction of $\Phi$ can incorporate *prior* information about $(q, \mathcal{T})$ (i.e., *inductive bias*) into the learning process. Throughout the paper, we assume that $q$ is *faithful* w.r.t. $\mathcal{T}$. That is, we aim to characterize the non-hallucinating learnability in the *cleanest* scenario where the entire training set is truthful. The main goal of this paper is to address the following *meta*-problem:

> *What structural assumptions on the tuple $(q, \mathcal{T})$ are necessary to achieve non-hallucinating learnability, even when the training set is entirely truthful?*

**Proper v.s. Improper Learners.** We distinguish two learning paradigms – proper and improper learning. Let $\mathcal{P} \subset \Delta(\mathcal{X})$ be a *hypothesis* class; for instance, $\mathcal{P}$ may represent the class of all distributions specified by a transformer architecture with different parameters. We say a learning rule $\Phi$ is *proper* w.r.t. $\mathcal{P}$ if $\mathrm{img}(\Phi) \subset \mathcal{P}$, that is the learner must produce a model *within* the hypothesis class. In the proper learning case, learnability also depends on the structure of $\mathcal{P}$. Otherwise, we say that the leaner $\Phi$ is *improper*; i.e., its outcome is completely *unconstrained*, or equivalently, we take $\mathcal{P} := \Delta(\mathcal{X})$. We demonstrate in this paper that the characterization of proper and improper learnability are fundamentally different for non-hallucinate learning of generative models.

We now summarize our main contributions as follows:

---

[1]Although in practice the model may also depend on certain *prompts*, this can be incorporated via *conditional* sampling. For clarity of exposition, we consider this simplified version as in Kalai & Vempala (2024).

- **Agnostic proper learning is impossible.** Clearly, a natural question one may ask is why do we need assumptions on the tuple $(q, \mathcal{T})$ at all. This is a valid argument; for instance, in the context of PAC learning, we do not need to make any specific assumptions on the data generation process. Instead, we quantify the goodness of the learner by comparing its performance to the best hypothesis in a hypothesis class $\mathcal{P}$. Learnability then reduces to characterizing the complexity of $\mathcal{P}$. We show in Theorem 1 that, perhaps surprisingly, such an *agnostic* competitive guarantee is *impossible* for non-hallucinating learning, even for a hypothesis class with only two elements, if the learner is proper. Our proof follows a probabilistic construction of hard instances of $(q, \mathcal{T})$, which also applies to other natural variants of non-hallucinating learning and is of independent interest.

- **Improper learning is possible and generalizes under VC concept classes.** Our second main result is a characterization of non-hallucinating learnability with *improper* learners. It is easy to observe that a naïve improper learner that produces the *empirical* distribution over the training set $\mathbf{x}^n$ never hallucinates (provided the demonstrator is faithful). This is clearly undesirable, since the produced model does not *generalize*. To resolve this issue, we introduce a new measure of *generalizability*, by requiring that the model contain as much *information* as the demonstrator, in addition to being non-hallucinating. We show in Theorem 2 that if the facts set $\mathcal{T}$ lies in a *concept* class $\mathcal{C}$ of finite VC-dimension, then non-hallucinating is possible with an improper learner that also generalizes to the amount of information of the demonstrator. We further show in Theorem 3 a matching sample complexity lower bound on the VC-dimension for the *worst-case* concept classes.

- **Proper learning can be hard even with VC concept class.** Finally, we address the case of non-hallucinating learning with *proper* learners. We show in Example 3 a hypothesis class $\mathcal{P}$ and concept class $\mathcal{C}$ that are of size two, yet any *proper* learner must hallucinate. We then identify a sufficient condition on the triplet $(q, \mathcal{C}, \mathcal{P})$ in Theorem 4 that allows non-hallucinating learning with proper learner, which is also necessary for certain cases.

**Interpretation.** Our results demonstrate that non-hallucinating learning is statistically *impossible* by solely leveraging the training data set at hand, even if it is completely truthful. Therefore, to reduce hallucination in the learned model, one must incorporate certain *inductive biases* that depend on the facts set itself. This aligns with practical approaches, such as the *post-training* process that incorporates human feedback (Achiam et al., 2023). Our work provides a systematic approach to achieving this by restricting the facts set to a *concept* class of finite VC-dimension. Although our result is primarily conceptual, it serves as the first step towards a *principled* approach to addressing hallucination in generative models.

## 1.1 RELATED WORK

Our work is related to the recent theoretical investigation in Kalai & Vempala (2024), which demonstrates that under certain regularity conditions, a model that is well *calibrated* must hallucinate, highlighting the importance of incorporating "factual information" into the learning process. Other impossibility results have also been reported very recently from a *computability* point of view, such as Xu et al. (2024); Banerjee et al. (2024). While our impossibility results are in the same spirit, our primary focus is to characterize the *minimal* assumptions under which non-hallucinating learning is statistically *possible*. Moreover, our formulation and results are grounded in learning theory (Shalev-Shwartz & Ben-David, 2014), where we treat the facts set itself as a research object, rather than adding ad-hoc constraints. This offers more flexibility in incorporating *domain-specific* factual constraints, instead of resolving hallucination in a "universal" sense. There has been extensive empirical investigation on understanding and addressing hallucinations, such as factual data enhancement (Dziri et al., 2022), hallucination detection (Manakul et al., 2023), and fine-tuning with human feedback (Ouyang et al., 2022). We refer to surveys by Huang et al. (2023); Ji et al. (2023); Zhang et al. (2023) and the references in Kalai & Vempala (2024) for more extensive discussions. Our work complements these investigations by providing a *principled view* aimed at understanding hallucination in learning generative models. Technically, our formulation of non-hallucinating learnability is related to *distribution PAC learning*, as investigated in Ashtiani et al. (2020); Bousquet et al. (2019; 2021). Our lower bounding techniques are rooted in information-theoretical methods, such as Le Cam's two-point method and Fan's inequality (Polyanskiy & Wu, 2022).

## 2 PRELIMINARIES

We review in this section some notations and concepts from the classical *distribution learning* literature, and highlight some immediate connections with our *non-hallucinating* learning paradigm. Let $\mathcal{X}$ be an instance space, and $\mathcal{P} \subset \Delta(\mathcal{X})$ be a hypothesis class. There exists some (unknown) ground truth distribution $q \in \Delta(\mathcal{X})$ (not necessarily in $\mathcal{P}$) that generates a sample $\mathbf{x}^n$ i.i.d. from $q$. The distribution *Probably Approximately Correct (PAC)* learning problem aims to find a learning rule $\Phi$ so that the produced distribution $\hat{p}_n := \Phi(\mathbf{x}^n)$ satisfies (for some $\alpha, \epsilon > 0$):

$$\|q - \hat{p}_n\|_{\mathsf{TV}} \leq \alpha \inf_{p \in \mathcal{P}} \|q - p\|_{\mathsf{TV}} + \epsilon, \tag{3}$$

with probability $\geq 1 - \delta$ over $\mathbf{x}^n \sim q$, where $\|\cdot\|_{\mathsf{TV}}$ denotes for total variation distance.

A hypothesis class $\mathcal{P}$ is said to be $\alpha$-agnostic PAC learnable if there exists a learning rule $\Phi$ such that for any $\epsilon, \delta > 0$ there exists a number $n$ (sample size) such that, for *any* distribution $q$, equation (3) holds. It can be shown (see e.g. Bousquet et al. (2022)) that *any finite* hypothesis class $\mathcal{P}$ is 3-agnostic learnable for *proper* learners. Intuitively, agnostic (proper) PAC learning ensures that the learned distribution's total variation distance to the ground truth $q$ is as close as possible to the best hypothesis in $\mathcal{P}$ that minimizes this distance.

For the *non-hallucinating* learning paradigm investigated in this paper, we need additionally an (unknown) *facts set* $\mathcal{T}$ such that the produced model $\hat{p}_n$ has a small hallucination rate $\mathsf{hall}(\hat{p}_n, \mathcal{T})$ as in (1). Clearly, if $q$ is *realizable* w.r.t. $\mathcal{P}$, i.e., $\inf_{p \in \mathcal{P}} \|q - p\|_{\mathsf{TV}} = 0$, then $\alpha$-agnostic PAC learning implies non-hallucinating learning, as demonstrated in the following fact:

**Fact 1** (Realizable learning of $q$ implies non-hallucinate learning)**.** *For any facts set $\mathcal{T}$ and any faithful demonstrator $q$ w.r.t. $\mathcal{T}$, if there exists a learner $\Phi$ such that with probability (w.p.) $\geq 1 - \delta$ over $\mathbf{x}^n \sim q$ we have $\|\Phi(\mathbf{x}^n) - q\|_{\mathsf{TV}} \leq \epsilon$, then w.p. $\geq 1 - \delta$ over $\mathbf{x}^n \sim q$ the following holds:*

$$\mathsf{hall}(\Phi(\mathbf{x}^n), \mathcal{T}) \leq \epsilon.$$

*Proof.* We have $\|\Phi(\mathbf{x}^n) - q\|_{\mathsf{TV}} = \sup_{A \subset \mathcal{X}} |\Phi(\mathbf{x}^n)[A] - q[A]| \leq \epsilon$. Therefore, taking $A := \bar{\mathcal{T}}$ we have $\mathsf{hall}(\Phi(\mathbf{x}^n), \mathcal{T}) \leq q[\bar{\mathcal{T}}] + \epsilon = \epsilon$, since $q[\bar{\mathcal{T}}] = \mathsf{hall}(q, \mathcal{T}) = 0$ due to faithfulness. $\qquad\square$

Fact 1 demonstrates that if the learned distribution is $\epsilon$-close to a faithful demonstrator $q$ under total variation distance, then $\Phi$ is guaranteed to be non-hallucinating with a hallucination rate upper bounded by $\epsilon$. At first glance, this may suggest that non-hallucinating learning is easier than distribution PAC learning. However, it is important to note that the requirement for the learned model to be close to the *ground truth* distribution $q$ is rather strong. This necessitates that: (i) the model class correctly approximates the ground truth distribution, and (ii) there are sufficient training samples to enable the learned model to generalize. Neither of these requirements is guaranteed or verifiable in practice, since the ground truth distribution $q$ is *unknown*.

In fact, as we will show in the following sections, (agnostic) learning of the demonstration distribution $q$ is neither sufficient nor necessary to achieve non-hallucinating learning.

## 3 IMPOSSIBILITY RESULTS

In this section, we establish several *impossibility* results for certain natural formulations of *agnostic* (proper) non-hallucinating learning. Analogous to the $\alpha$-agnostic (distribution) PAC learning formulation, one may consider the following definition:

**Definition 1** ($\alpha$-agnostic non-hallucinating learning)**.** *A hypothesis class $\mathcal{P} \subset \Delta(\mathcal{X})$ is $\alpha$-agnostic non-hallucinating learnable if for some $\alpha > 0$ there exists a proper learner $\Phi$ such that for any given $\epsilon, \delta > 0$, there exists a number $n$, such that for any facts set $\mathcal{T}$ and faithful-demonstrator $q$ w.r.t. $\mathcal{T}$*

$$\Pr_{\mathbf{x}^n \overset{i.i.d.}{\sim} q} \left[ \mathsf{hall}(\Phi(\mathbf{x}^n), \mathcal{T}) - \alpha \inf_{p \in \mathcal{P}} \mathsf{hall}(p, \mathcal{T}) \geq \epsilon \right] \leq \delta.$$

Intuitively, $\alpha$-agnostic non-hallucinating learnability requires that with high probability (w.h.p.), the learner produce a distribution within the class $\mathcal{P}$ that has hallucination rate not much higher than

the minimal achievable hallucination rate of $\mathcal{P}$, *regardless* of how the facts set $\mathcal{T}$ and demonstrator $q$ are chosen. It is crucial to note that we do not require the learned distribution to be close to the demonstrator $q$ under total variation. Although this may appear to be a much weaker condition than learning the distribution $q$ itself, the following example demonstrates that it is *impossible* even for a hypothesis class with two elements and the minimal achievable hallucination rate is 0.

**Example 1.** *Let $A$, $A_1$, and $A_2$ be disjoint non-empty sets. We define $p_1$ and $p_2$ to be any distributions supported over $A_1$ and $A_2$, respectively. Take $\mathcal{P} := \{p_1, p_2\}$ and $q$ to be any distribution supported over $A$. Now, for any given learner $\Phi$ and sample size $n$, $\Phi$ must produce $\hat{p}_n \in \mathcal{P}$. Therefore, w.p. $\geq \frac{1}{2}$ over the sample $\mathbf{x}^n \sim q$, it will select $\hat{p}_n := p_i$ for some $i \in \{1, 2\}$. We now (adversarially) select the facts set $\mathcal{T} := A \cup A_{3-i}$. It can be verified that: (1) $q$ is faithful w.r.t. $\mathcal{T}$; (2) $\mathsf{hall}(\hat{p}_n, \mathcal{T}) = 1$; and (3) $\inf_{p \in \mathcal{P}} \mathsf{hall}(p, \mathcal{T}) = 0$. This implies that the learner $\Phi$ cannot satisfy Definition 1 for all $\epsilon < 1$, $\delta < \frac{1}{2}$ and $\alpha > 0$. Since the learner $\Phi$ is selected arbitrarily, the class $\mathcal{P}$ is* not *$\alpha$-agnostic non-hallucinating learnable for all $\alpha > 0$.*

We mentioned in Section 2 that any *finite* hypothesis class is 3-agnostic (proper) PAC learnable but Example 1 demonstrates a striking distinction between the agnostic distribution PAC learning and the agnostic *non-hallucinating* learning.

One may noticed that Example 1 is fairly pathological, since the constructed distribution $q$ is supported *outside* of the support of $p_1$ and $p_2$. Therefore, in effect, it provides no information for distinguishing between $p_1$ and $p_2$. For this reason, we introduce the following refined definition of hallucination rate *relative* to $q$.

**Definition 2** (Relative Hallucination Rate). *For any given distributions $p, q$, the $\epsilon$-relative hallucination rate of $p$ w.r.t. $q$ is defined as:*

$$\mathsf{hall}_\epsilon(p, q) = \sup_{A \subset \mathcal{X}, q[A] \leq \epsilon} p[A].$$

Intuitively, the *relative* hallucination rate $\mathsf{hall}_\epsilon(p, q)$ measures the maximum probability that $p$ assigns to a set where the distribution $q$ has a small mass. In other words, the distribution $p$ is non-hallucinating relative to $q$ if every *rare* event under $q$ is also rare under $p$. Observe that, if $q$ is faithful w.r.t. some facts set $\mathcal{T}$, then:

$$\forall \epsilon \geq 0, \ \mathsf{hall}(p, \mathcal{T}) \leq \mathsf{hall}_\epsilon(p, q). \tag{4}$$

Therefore, a small relative hallucination rate automatically implies small (absolute) hallucination rate as in (1). However, the converse is generally not true. To see this, consider Example 1, we know that $\inf_{p \in \mathcal{P}} \mathsf{hall}(p, \mathcal{T}) = 0$ while $\inf_{p \in \mathcal{P}} \mathsf{hall}_\epsilon(p, q) = 1$ for all $\epsilon \geq 0$, thus the impossibility result implied therein no longer holds. It is interesting to note that the relative hallucination rate can be naturally bounded by the notion of $\sigma$-smoothness (Haghtalab et al., 2022). We say $p$ is $\sigma$-smooth w.r.t. $q$ if $\forall A \subset \mathcal{X}$ we have $p[A] \leq \frac{1}{\sigma} q[A]$, therefore, $\mathsf{hall}_\epsilon(p, q) \leq \frac{\epsilon}{\sigma}$ for all $\epsilon \geq 0$.

We show in the following theorem that $\alpha$-agnostic-non-hallucinating learning is *impossible* for all given $\alpha > 0$ even for the *relative* hallucination rate and for a hypothesis class of size 2.

**Theorem 1** (Agnostic proper non-hallucinating learning is impossible). *There exists a hypothesis class $\mathcal{P}$ of size 2 such that for* any proper *learning rule $\Phi$, parameter $\delta \leq \frac{1}{3}$, and any sample size $n$, there exists a tuple $(q, \mathcal{T})$ such that for all $\epsilon < \frac{1}{2}$, with probability $> \delta$ over $\mathbf{x}^n \sim q$:*

1. *$q$ is faithful w.r.t. $\mathcal{T}$;*

2. *$\inf_{p \in \mathcal{P}} \mathsf{hall}(p, \mathcal{T}) \leq \inf_{p \in \mathcal{P}} \mathsf{hall}_\epsilon(p, q) \leq 2\epsilon$, i.e., there exists non-hallucinating $p \in \mathcal{P}$;*

3. *$\mathsf{hall}_\epsilon(\Phi(\mathbf{x}^n), q) \geq \mathsf{hall}(\Phi(\mathbf{x}^n), \mathcal{T}) = 1$, i.e., the learned model hallucinates.*

*Proof.* Let $p_1$ be the uniform distribution over $A_1 := [0, 0.5]$, $p_2$ be uniform over $A_2 := [0.5, 1]$ and $\mathcal{P} = \{p_1, p_2\}$, where $[a, b]$ denotes the *interval* in the real line from $a$ to $b$. We now describe a way of selecting $(q, \mathcal{T})$ satisfying the conditions in the theorem statement. To do so, we use a *probabilistic* argument. Let $\mu_1$ and $\mu_2$ be two distributions over the tuple $(q, \mathcal{T})$ such that:

1. For distribution $\mu_1$, we select the facts set $\mathcal{T} := A_1 \cup \tilde{A}_2$ where $\tilde{A}_2$ is (a discrete set) sampled *i.i.d.* uniformly from $A_2$ with $|\tilde{A}_2| \gg 2n^2$. After generating $\mathcal{T}$, we take $q = \frac{1}{2}\mathsf{Uni}(A_1) + \frac{1}{2}\mathsf{Uni}(\tilde{A}_2)$, i.e., w.p. $\frac{1}{2}$ uniform over $A_1$ and w.p. $\frac{1}{2}$ uniform over $\tilde{A}_2$;

2. For distribution $\mu_2$, we select the facts set $\mathcal{T} := A_2 \cup \tilde{A}_1$ where $\tilde{A}_1$ is (a discrete set) sampled *i.i.d.* uniformly from $A_1$ with $|\tilde{A}_1| \gg 2n^2$. After generating $\mathcal{T}$, we take $q = \frac{1}{2}\mathsf{Uni}(A_2) + \frac{1}{2}\mathsf{Uni}(\tilde{A}_1)$ interpreted as above.

By construction, we know that the demonstrator $q$ is completely faithful w.r.t. $\mathcal{T}$, and for any $\epsilon < \frac{1}{2}$, the following key properties hold:

1. If $(q, \mathcal{T}) \sim \mu_1$ then $\mathsf{hall}(p_1, \mathcal{T}) \le \mathsf{hall}_\epsilon(p_1, q) \le 2\epsilon$ and $\mathsf{hall}_\epsilon(p_2, q) \ge \mathsf{hall}(p_2, \mathcal{T}) = 1$;

2. If $(q, \mathcal{T}) \sim \mu_2$ then $\mathsf{hall}(p_2, \mathcal{T}) \le \mathsf{hall}_\epsilon(p_2, q) \le 2\epsilon$ and $\mathsf{hall}_\epsilon(p_1, q) \ge \mathsf{hall}(p_1, \mathcal{T}) = 1$.

Therefore, the first two conditions in the theorem statement are satisfied.

We take now any $\delta \le \frac{1}{3}$ and assume for now that condition 3 *does not* hold. We have if $(q, \mathcal{T}) \sim \mu_1$ then $\Phi(\mathbf{x}^n) = p_1$ happens w.p. $\ge 1 - \delta \ge \frac{2}{3}$ over the randomness $\mathbf{x}^n \sim q$; else if $(q, \mathcal{T}) \sim \mu_2$ then $\Phi(\mathbf{x}^n) = p_1$ happens w.p. $\le \delta \le \frac{1}{3}$ over the randomness $\mathbf{x}^n \sim q$ (recall that the learner $\Phi$ must output either $p_1$ or $p_2$ since it is assumed to be *proper*). For $i \in \{1, 2\}$, we define the *mixture distribution* $\nu_i$ over $\mathcal{X}^n$ as described by the outcome of the following Markov process:

$$\mu_i \sim (q, \mathcal{T}) \overset{i.i.d. \text{ from } q}{\sim} \mathbf{x}^n.$$

Therefore, we have

$$\|\nu_1 - \nu_2\|_{\mathsf{TV}} = \sup_{E \subset \mathcal{X}^n} \nu_1(E) - \nu_2(E) \ge \nu_1(E') - \nu_2(E') \ge \frac{2}{3} - \frac{1}{3} = \frac{1}{3},$$

where $E' := \{\mathbf{x}^n \in \mathcal{X}^n : \Phi(\mathbf{x}^n) = p_1\}$ and the second inequality follows by discussion above.

Note that, the randomness of $\nu_i$ has two parts: w.p. $\frac{1}{2}$ the sample is uniform over $A_i$ and w.p. $\frac{1}{2}$ the sample is uniform over $\tilde{A}_{3-i}$. The set $\tilde{A}_{3-i}$ is selected uniformly from $A_{3-i}$ and is of size $\gg 2n^2$. Therefore, conditioning on the event that there is *no repetition* in sample $\mathbf{x}^n$, the distribution of $\mathbf{x}^n \sim \nu_i$ restricted on $A_{3-i}$ is *exactly* the same as sampling *i.i.d.* uniformly from $A_{3-i}$. Let $E$ be the event over $\mathcal{X}^n$ so that $\mathbf{x}^n \in E$ has no repetition. We have:

$$\|\nu_1(\cdot \mid E) - \nu_2(\cdot \mid E)\|_{\mathsf{TV}} = 0.$$

Since we have selected the size $|\tilde{A}_i| \gg 2n^2$ for $i \in \{1, 2\}$, by the *birthday paradox* (Katz & Lindell, 2007, Lemma A.9), we have $\nu_1(E) = \nu_2(E) \ge 1 - \frac{1}{4} = \frac{3}{4}$. This implies that:

$$\begin{aligned}
\|\nu_1 - \nu_2\|_{\mathsf{TV}} &= \nu_1(B) - \nu_2(B), \text{ for some } B \subset \mathcal{X}^n \\
&= \nu_1(B \cap E) - \nu_2(B \cap E) + \nu_1(B \cap \bar{E}) - \nu_2(B \cap \bar{E}) \\
&= \nu_1(E)(\nu_1(B \mid E) - \nu_2(B \mid E)) + \nu_1(\bar{E})(\nu_1(B \mid \bar{E}) - \nu_2(B \mid \bar{E})), \text{ since } \nu_1(E) = \nu_2(E) \\
&\le \nu_1(E)\|\nu_1(\cdot \mid E) - \nu_2(\cdot \mid E)\|_{\mathsf{TV}} + \nu_1(\bar{E})\|\nu_1(\cdot \mid \bar{E}) - \nu_2(\cdot \mid \bar{E})\|_{\mathsf{TV}} \\
&\le \nu_1(\bar{E}) < 1 - \frac{3}{4} < \frac{1}{3}.
\end{aligned}$$

This contradicts to our previous conclusion that $\|\nu_1 - \nu_2\|_{\mathsf{TV}} \ge \frac{1}{3}$, thus the presumption that condition 3 does not hold is *not true*. Therefore, proof of theorem is completed. $\square$

Note that, although Theorem 1 is proved only for a specifically constructed pair of distributions, the arguments can be extended to *any* pair of distributions that have sufficiently separated densities, provided the distributions are smooth enough to allow for sufficiently long *non-repetitive* samples.

Theorem 1 implies the following corollary that strengthens Example 1, which shows that agnostic (proper) non-hallucinating learning can be *harder* than agnostic PAC learning, even if the hypothesis class is of size 2 and even there exists a hypothesis within class that is *not* (relatively) hallucinating.

**Corollary 1.** *There exists a hypothesis class $\mathcal{P}$ of size 2, such that $\mathcal{P}$ is not $\alpha$-agnostic non-hallucinating learnable for any given $\alpha \ge 0$. This is true for both the (absolute) hallucinate rate as defined in (1) and the relative hallucination rate in Definition 2.*

*Proof.* We only prove the case for the *relative* hallucination rate, as the case for (absolute) hallucination rate follows similarly. For any given $\alpha > 0$, we take $\epsilon$ small enough so that $\epsilon \leq \frac{1}{2\alpha+1}$. By Theorem 1 condition 2, we have $\inf_p \mathsf{hall}_\epsilon(p, q) \leq 2\epsilon$. Moreover, by condition 3 of Theorem 1, we have w.p. $> \delta$ that $\mathsf{hall}_\epsilon(\Phi(\mathbf{x}^n), q) = 1$. Therefore, we conclude w.p. $> \delta$ that:

$$\mathsf{hall}_\epsilon(\Phi(\mathbf{x}^n), q) - \alpha \inf_{p \in \mathcal{P}} \mathsf{hall}_\epsilon(p, q) \geq 1 - \alpha \cdot 2\epsilon \geq \epsilon$$

whenever $\epsilon \leq \frac{1}{2\alpha+1}$. This violates the $\alpha$-agnostic-non-hallucinating learnability in Definition 1. $\square$

# 4 NON-HALLUCINATING LEARNING VIA KNOWLEDGE OF $\mathcal{T}$

As demonstrated in Theorem 1, agnostic (proper) non-hallucinating learning is *impossible* by restricting the *hypothesis class* alone, even when measured *competitively* (i.e., by comparing with the best model in a hypothesis class). This is partially due to the fact that the learner must handle *any* tuple $(q, \mathcal{T})$, or in other words, it does not incorporate any *prior* knowledge of the facts set into the learning process. This phenomenon mirrors the "no-free lunch" theorems in the PAC learning literature (Shalev-Shwartz & Ben-David, 2014), which assert that learning is *impossible* without assumptions about the learning target. This is typically resolved by introducing additional *inductive biases* in the learning process that restrict the hypothesis class, such as limiting it to a finite VC-dimension. This section is devoted to introducing certain natural assumptions on $(q, \mathcal{T})$ that lead to non-trivial resolutions of non-hallucinating learning.

## 4.1 THE IMPROPER LEARNING CASE

Clearly, if the leaner is *improper*, then a naïve non-hallucinating learner that simply generates the *empirical distribution* over the training sample $\mathbf{x}^n$ never hallucinates. This is clearly not satisfactory, as the learned model is completely non-generalizable. To mitigate this restriction, we introduce certain constrains on the facts set $\mathcal{T}$ that allow our learned model to "generalize". We assume that $\mathcal{T} \in \mathcal{C}$, where $\mathcal{C} \subset 2^\mathcal{X}$ is a *concept class* of all possible sets that $\mathcal{T}$ can be chosen from. Here, the concept class would be determined by *prior* knowledge of the learner on $\mathcal{T}$.

To quantify the "generalizability" of the learned model, we introduce an additional notion of *information measure I* that quantifies the amount of "information" of distributions in $\Delta(\mathcal{X})$:

$$I : \Delta(\mathcal{X}) \times \mathcal{X}^* \to \mathbb{R}^{\geq 0}.$$

Note that, here, we allow the information measure $I(p, \mathbf{x}^n)$ to be dependent on the training set $\mathbf{x}^n$ as well. For any given information measure function $I$, we consider the following definition:

**Definition 3** (Improper non-hallucinating learnable). *For any $\gamma < 1$, a concept class $\mathcal{C}$ is said to be $\gamma$-approximately improper non-hallucinating learnable w.r.t. information measure $I$, if there exists an (improper) learner $\Phi$ such that for any $\epsilon, \delta > 0$, there exists a number $n$, so that for any $\mathcal{T} \in \mathcal{C}$ and any faithful-demonstrator $q$ over $\mathcal{T}$, w.p. $\geq 1 - \delta$ over $\mathbf{x}^n \sim q$ the following holds:*

1. *The hallucination rate $\mathsf{hall}(\Phi(\mathbf{x}^n), \mathcal{T}) \leq \epsilon$;*

2. *$I(\Phi(\mathbf{x}^n), \mathbf{x}^n) \geq (1 - \gamma)I(q, \mathbf{x}^n)$.*

Observe that the first part of Definition 3 ensures that the learned model does not hallucinate, while the second part ensures that the model is as informative as the demonstrator so that generalizability is possible. Moreover, it is crucial to note that, although the facts set $\mathcal{T}$ is restricted to the concept class $\mathcal{C}$, the demonstrator $q$ is *unconstrained*, as long as it is faithful w.r.t. $\mathcal{T}$.

There are many natural information measures. We mention a few here:

1. For discrete $\mathcal{X}$, *Shannon entropy* is defined as $H(p) := \sum_{\mathbf{x} \in \mathcal{X}} p[\mathbf{x}] \log \frac{1}{p[\mathbf{x}]}$, which measures the (absolute) information contained in $p$ *independent* of the training set $\mathbf{x}^n$;

2. The *$\alpha$-Rényi entropy* is defined as $H_\alpha(p) = \frac{1}{1-\alpha} \log \left( \sum_{\mathbf{x} \in \mathcal{X}} p[\mathbf{x}]^\alpha \right)$, where $\alpha > 0$ and $\alpha \neq 1$, which subsumes the Shannon entropy for $\alpha \to 1$;

3. The *out-of-sample mass* is defined as (for $n \geq 1$): $I(p, \mathbf{x}^n) := p[\mathcal{X} \backslash \{\mathbf{x}_1, \cdots, \mathbf{x}_n\}]$, which measures the amount of probability mass of $p$ assigned *outside* of the training set.

The specific choice of information measure typically depends on the task at hand. We show in the following theorem, perhaps surprisingly, that improper non-hallucinating learning is possible if the concept class $\mathcal{C}$ has finite VC-dimension, *regardless* of what information measure is selected.

**Theorem 2.** *If a concept class $\mathcal{C}$ has finite VC-dimension, then $\mathcal{C}$ is $0$-approximately (improper) non-hallucinating learnable w.r.t.* any *information measure function $I$. Moreover, the sample complexity is upper bounded by* $n \leq O\left(\frac{\mathsf{VC}(\mathcal{C})\log(1/\epsilon) + \log(1/\delta)}{\epsilon}\right)$.

*Proof.* Let $\mathcal{T}^* \in \mathcal{C}$ be a ground truth facts set and $q^*$ be a faithful-demonstrator w.r.t. $\mathcal{T}^*$. Let $\mathbf{x}^n \sim q^*$ be a set of samples. We define the *version space*:

$$\mathcal{C}_n = \{\mathcal{T} \in \mathcal{C} : \forall i \leq n, \ \mathbf{x}_i \in \mathcal{T}\}, \tag{5}$$

as the subset of elements in $\mathcal{C}$ that are consistent with the sample $\mathbf{x}^n$. Note that $\mathcal{T}^* \in \mathcal{C}_n$ holds always, since $q^*$ is faithful. For any $\mathcal{T} \in \mathcal{C}$, we say $q^*$ $\epsilon$-violates $\mathcal{T}$ if $\mathsf{hall}(q^*, \mathcal{T}) \geq \epsilon$. We claim that:

$$\Pr_{\mathbf{x}^n \sim q^*}[\exists \mathcal{T} \in \mathcal{C}_n, \ s.t. \ q^* \ \epsilon\text{-violates } \mathcal{T}] \leq \delta, \tag{6}$$

provided $n \geq C\frac{d\log(1/\epsilon) + \log(1/\delta)}{\epsilon}$ for some constant $C$, where $d = \mathsf{VC}(\mathcal{C})$. To see this, we can view the sample $\mathbf{x}^n$ as feature-label pairs that has all labels being $1$. In this case, the hallucination rate $\mathsf{hall}(q^*, \mathcal{T})$ can be viewed as the population risk of $\mathcal{T}$ under the data distribution. Moreover, since $q^*$ is faithful, the ground truth set $\mathcal{T}^*$ achieves $0$ risk, therefore, the data distribution is effectively *realizable* w.r.t. $\mathcal{C}$. The claim then follows by standard uniform convergence result of VC class, see e.g. Shalev-Shwartz & Ben-David (2014, Theorem 6.8 (3)).

We now consider the following learning rule:

$$\Phi(\mathbf{x}^n) := \arg\max_{p \in \Delta(\mathcal{X})} \{I(p, \mathbf{x}^n) : p \text{ does not } \epsilon\text{-violate } \mathcal{T}, \ \forall \mathcal{T} \in \mathcal{C}_n\}. \tag{7}$$

Note that, $\Phi(\mathbf{x}^n)$ always satisfies condition 1 of Definition 3, since $\mathcal{T}^* \in \mathcal{C}_n$ and therefore any feasible solution from (7) does not $\epsilon$-violate $\mathcal{T}^*$. Moreover, by (6), w.p. $\geq 1 - \delta$ over $\mathbf{x}^n \sim q^*$ that $q^*$ is within the feasible set of (7). Therefore, we have:

$$I(\Phi(\mathbf{x}^n), \mathbf{x}^n) \geq I(q^*, \mathbf{x}^n),$$

by definition of $\arg\max$. Thus, condition 2 of Definition 3 is also satisfied with $\gamma = 0$. $\qquad\square$

Theorem 2 shows that if the facts set $\mathcal{T}$ can be specified by a neural network with binary outcome (which has finite VC-dimension), then an improper learner can yield a model that avoids hallucinations while contains as much information as the demonstrator. This is conceptually analogous to approaches in practice, such as Reinforcement Learning from Human Feedback (RLHF), where the reward model can be viewed as a "concept function" that characterizes the facts. However, our approach differs in that our "reward model" is learned solely from the training data without human input. The precise learning rule can be extracted from (7), albeit computationally inefficient.

It is crucial to distinguish our non-hallucinating learning paradigm in Theorem 2 from the standard PAC learning (Shalev-Shwartz & Ben-David, 2014), even though the proof may look similar. This is because, in our non-hallucinating learning framework, the ultimate goal is to produce a *distribution* (i.e., a generative model) not a classification function (i.e., a discriminative model). The concept class only serves as a *regularization* to control the hallucination rate of the learned model. More importantly, our learning paradigm is *unsupervised* without any need of human labeling.

We now establish a matching sample complexity lower bound for (improper) non-hallucinating learning w.r.t. the *out-of-sample mass* information measure for the *worst* concept classes $\mathcal{C}$. This, together with Theorem 2, implies that the VC-dimension remains a meaningful complexity measure that characterizes improper non-hallucinating learning.

**Theorem 3.** *For any $d \in \mathbb{N}$, there exists a concept class $\mathcal{C}$ of VC-dimension $d$ such that for any $\epsilon > 0$ sufficiently small and any $\delta \leq \epsilon$, the sample complexity of $0$-approximately (improper) non-hallucinating learning $\mathcal{C}$ w.r.t. out-of-sample mass measure is lower bounded by $\Omega(\frac{d}{\epsilon})$.*

*Sketch of Proof.* We only sketch the main idea here and refer to Appendix A for complete details. At a high level, our goal is to construct, for any $d \in \mathbb{N}$, a concept class $\mathcal{C}$ of VC-dimension $d$ that

achieves our claimed sample complexity lower bound. This is done by showing that if condition (2) of Definition 3 is satisfied for the *out-of-sample* information measure, then for any training set of size $\ll \frac{d}{\epsilon}$ and any learning rule, the *expected* hallucination rate must exceed $2\epsilon$ (this constitutes our main technical contribution). Therefore, condition (1) of Definition 3 will be violated when taking $\delta := \epsilon$. This implies that for any sample size $\ll \frac{d}{\epsilon}$, one cannot *simultaneously* satisfy both condition (1) and (2) of Definition 3, thus establishing the lower bound. □

Note that, although Theorem 3 is proved only for the *out-of-sample mass* measure, a similar lower bound can also be established for other information measures. We refer to Appendix B for a lower bound on the *Shannon entropy*. Moreover, Theorem 3 only characterizes the sample complexity in the *minimax* sense, i.e., for the *worst* case VC classes. Indeed, the sample complexity can be *lower* for certain specific VC-classes, as demonstrated in the example below.

**Example 2.** *Let* $\mathcal{X} = A \cup \{1, \cdots, d\}$ *where* $|A| \gg d$; *we define class* $\mathcal{C}$ *as the class of all subsets of* $\mathcal{X}$ *that contain* $A$. *We have* $\mathsf{VC}(\mathcal{C}) \geq d$. *However, an algorithm that always produces* uniform *distribution over* $A$ *has* $0$ *hallucination rate and the out-of-sample mass is lower bounded by* $\frac{|A|-n}{|A|}$ *for any sample of size* $n$. *Taking* $n = 1$, *one can make the out-of-sample mass arbitrarily close to* $1$ *by choosing sufficiently large* $|A|$.

We would like to point out that any complexity measure on $\mathcal{C}$ that characterizes the (improper) non-hallucinating learnability at the *instance* level would necessarily depend on the information measure (for the lower bounds). We leave it as an open problem to find more fine-grained characterizations tailored to specific information measures, such as for the out-of-sample mass and Shannon entropy.

## 4.2 THE PROPER LEARNING CASE

We demonstrated in Theorem 2 that non-hallucinating learning is possible for *improper* learners if the facts set lies in a concept class of finite VC-dimension. A natural questions is whether such an approach can be extended to the *proper* learning case, i.e., instead of allowing the learner to output an *arbitrary* distribution in $\Delta(\mathcal{X})$, we restrict the output to some *hypothesis class* $\mathcal{P} \subset \Delta(\mathcal{X})$.

We again assume that the facts set $\mathcal{T}$ is selected from some concept class $\mathcal{C}$. Let $q$ be any faithful demonstrator w.r.t. $\mathcal{T}$ and $\mathbf{x}^n \sim q$. A natural learning rule that is inspired from (7) is as follows:

$$\Phi(\mathbf{x}^n) = \arg\max_{p \in \mathcal{P}} \left\{ I(p, \mathbf{x}^n) : \mathsf{hall}(p, \mathcal{T}) \leq \epsilon, \ \forall \mathcal{T} \in \mathcal{C}_n \right\}, \tag{8}$$

where $\mathcal{C}_n$ is the *version space* defined in (5) and $I$ is some appropriately selected information measure. Unfortunately, this (in fact any proper) learning rule fails to achieve non-hallucinating learning even if the minimal achievable hallucination rate of distributions in $\mathcal{P}$ is $0$. This is discussed in the next example.

**Example 3.** *There exist hypothesis class* $\mathcal{P}$ *and concept class* $\mathcal{C}$ *that are of size* $2$, *such that for any* proper *learner* $\Phi$, *we can select some* $\mathcal{T} \in \mathcal{C}$ *and faithful demonstrator* $q$ *w.r.t.* $\mathcal{T}$ *such that* $\Pr_{\mathbf{x}^n \sim q} [\mathsf{hall}(\Phi(\mathbf{x}^n), \mathcal{T}) \geq 0.99] \geq \frac{1}{2}$. *The construction is similar to that of Example 1. To see this, we take two sets* $A_1$ *and* $A_2$ *of size* $|A_1| = |A_2| = 100$ *and* $|A_1 \cap A_2| = 1$. *The concept class* $\mathcal{C} = \{A_1, A_2\}$ *and the hypothesis class* $\mathcal{P} = \{\mathsf{Uni}(A_1), \mathsf{Uni}(A_2)\}$. *Furthermore, we select demonstrator* $q = \delta_{\mathbf{x}_0}$ *for the (single) element* $\mathbf{x}_0 \in A_1 \cap A_2$, *which is faithful for both* $A_1$ *and* $A_2$. *Now, for any learner* $\Phi$, *it must produce* $\mathsf{Uni}(A_i)$ *w.p.* $\geq \frac{1}{2}$ *for some* $i \in \{1, 2\}$. *To satisfy the claimed lower bound, the adversary then simply takes the actual facts set* $\mathcal{T}$ *to be* $A_{3-i}$.

Although the impossibility result of Example 3 is not technically surprising, since the demonstrator $q$ effectively produces no "information." It demonstrates a striking distinction between proper and improper learnability of non-hallucinating learning. Recall that, for improper learning, the learner can *adapt* to the amount of information provided by the demonstrator, yet still avoid hallucination.

Therefore, in order to obtain meaningful results for the proper learning case, one needs to control the behaviour of the demonstrator $q$ as well. To achieve this, we introduce a sufficient condition under which the learning rule from (8) achieves non-hallucinating learnability.

**Definition 4.** *A demonstrator* $q$ *is said to be sufficiently informative for a pair* $(\mathcal{C}, \mathcal{P})$ *if there exists a function* $\xi : [0, 1] \to [0, 1]$ *such that for any* $\epsilon > 0$

$$\inf_{p \in \mathcal{P}} \sup_{\mathcal{T} \in \mathcal{C}_{\xi(\epsilon)}} \mathsf{hall}(p, \mathcal{T}) \leq \epsilon, \tag{9}$$

where $\mathcal{C}_{\xi(\epsilon)} = \{\mathcal{T} \in \mathcal{C} : \Pr_{\mathbf{x} \sim q}[\mathbf{x} \notin \mathcal{T}] \leq \xi(\epsilon)\}$.

Intuitively, Definition 4 ensures that the pathological instance of Example 3 do not occur for a sufficiently small "local neighborhood" $\mathcal{C}_{\xi(\epsilon)}$ of $\mathcal{C}$ induced by $p$. Indeed, equipped with this condition, we can establish the following positive result.

**Theorem 4.** *Let $\mathcal{C}$ be any concept class of finite VC-dimension and $\mathcal{P}$ be any hypothesis class. Then for any $\mathcal{T} \in \mathcal{C}$ and any sufficiently informative faithful-demonstrator $q$ w.r.t. $(\mathcal{C}, \mathcal{P})$ as in Definition 4, we have w.p. $\geq 1 - \delta$ over $\mathbf{x}^n \sim q$ that for the predictor $\Phi$ in (8) satisfies*

$$\Pr_{\mathbf{x}^n \sim q}[\mathsf{hall}(\Phi(\mathbf{x}^n), \mathcal{T}) \geq \epsilon] \leq \delta,$$

*provided $n \geq \Omega\left(\frac{\mathsf{VC}(\mathcal{C})\log(1/\xi(\epsilon)) + \log(1/\delta)}{\xi(\epsilon)}\right)$, where $\xi(\cdot)$ is the function in Definition 4.*

*Sketch of Proof.* The proof essentially follows a similar path as Theorem 2. At a high level, our goal is to show that, for sufficiently long samples of size $n$, the version space $\mathcal{C}_n$ as in (6) is contained within the "good" neighborhood $\mathcal{C}_{\xi(\epsilon)}$. Therefore, any distribution $p$ in the feasible solution of (8) must satisfy, for all $\mathcal{T} \in \mathcal{C}_n$, that $\mathsf{hall}(p, \mathcal{T}) \leq \epsilon$ due to Definition 4. The final hallucination rate bound then follows from the fact that the ground truth facts set is within $\mathcal{C}_n$. □

Note that, the condition in Definition 4 is, in a sense, the *minimal* requirement to achieve *proper* non-hallucinating learnability. Otherwise, consider any demonstrator $q$ that does not satisfy the condition. There must exist some $\epsilon > 0$ so that for any $\xi > 0$ we have:

$$\inf_{p \in \mathcal{P}} \sup_{\mathcal{T} \in \mathcal{C}_\xi} \mathsf{hall}(p, \mathcal{T}) > \epsilon.$$

A similar argument as in Example 3 yields that for any proper learner, there must exist some $\mathcal{T} \in \mathcal{C}_\xi$ so that the hallucination rate $> \epsilon$ w.p. $\geq \frac{1}{|\mathcal{P}|}$, provided $|\mathcal{C}_\xi| < \infty$ and $\xi$ is small enough.

From a practical point of view, condition (9) cannot be verified directly, as it depends on the *unknown* demonstrator $q$. However, as we demonstrated in the proof of Theorem 4, one only needs to verify the following condition

$$\inf_{p \in \mathcal{P}} \sup_{\mathcal{T} \in \mathcal{C}_n} \mathsf{hall}(p, \mathcal{T}) \leq \epsilon,$$

to ensure that the hallucination rate claimed in Theorem 4 holds. This can be verified *empirically* since $\mathcal{C}_n$ depends solely on the training data $\mathbf{x}^n$ and not on the actual distribution $q$.

## 5 Conclusion and Discussion

In this paper, we investigated the learnability of non-hallucinating generative models from a learning-theoretic perspective. We showed that agnostic non-hallucinating learning is statistically *impossible* if one does not incorporate knowledge of the ground truth facts set into the learning process, even when hallucination is measured *relatively*. This contrasts substantially with classical distribution PAC learning, where agnostic learning is possible without restrictions on the ground truth distribution. We then established several *positive* results, showing that non-hallucinating learning is achievable by restricting the ground truth facts set to certain *concept classes* with finite VC-dimension, and determined the tight sample complexity required to achieve learnability.

Although our contribution is primarily conceptual, we believe it will help practitioners understand the *fundamental limitations* of hallucinations in generative models in a more *principled* way. Our work also provides several algorithmic approaches for incorporating knowledge of the ground truth facts set into the learning process via *regularization*, such as the learning rules provided in (7) and (8), albeit computationally inefficiently. Our framework lays the foundation for significant future research on mitigating hallucinations in generative models. This includes exploring different approaches to incorporating factual knowledge, such as through logical rules, synthetic data generation, or leveraging human feedback, as well as developing more computationally efficient algorithms.

**Acknowledgments.** This work is partially supported by the NSF Center for Science of Information (CSoI) Grant CCF-0939370, and also by NSF Grants CCF-2006440 and and CCF-2211423.

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

## A  PROOF OF THEOREM 3

For any $d \in \mathbb{N}$, we take $\mathcal{X}$ to be a set of size $2d + 1$. Let $\mathbf{x}_0 \in \mathcal{X}$ be any element. We define $\mathcal{C} := \{\mathcal{T} \subset \mathcal{X} : \mathbf{x}_0 \in \mathcal{T} \text{ and } |\mathcal{T}| = 1 + d\}$. It is easy to verify that $\mathsf{VC}(\mathcal{C}) = d$. We now describe a random process for generating $(q, \mathcal{T})$. We first uniformly sample $\mathcal{T}$ from $\mathcal{C}$ and then define $q = (1 - \epsilon')\delta_{\mathbf{x}_0} + \epsilon'\mathsf{Uni}(\mathcal{T}\backslash\{\mathbf{x}_0\})$ for some $\epsilon'$ to be determined. Clearly, $q$ is faithful w.r.t. $\mathcal{T}$. Let $\mu$ be the derived distribution over $(q, \mathcal{T})$. For any given learner $\Phi$ and sample size $n \leq \frac{d}{4\epsilon'}$, we denote $\hat{p}_n := \Phi(\mathbf{x}^n)$. Our goal is to lower bound the expected hallucination rate:

$$\mathbb{E}_{(q,\mathcal{T})\sim\mu}\mathbb{E}_{\mathbf{x}^n\sim q}[\mathsf{hall}(\hat{p}_n, \mathcal{T})] = \mathbb{E}_{\mathbf{x}^n}\mathbb{E}_{(q,\mathcal{T})\sim\mu|\mathbf{x}^n}[\mathsf{hall}(\hat{p}_n, \mathcal{T})],$$

where $\mathbb{E}_{\mathbf{x}^n}$ is over the *mixture* $\mu \sim q \sim \mathbf{x}^n$ and $\mu \mid \mathbf{x}^n$ is the distribution of $\mu$ conditioning on $\mathbf{x}^n$.

Denote $A := \{\mathbf{x}_1, \cdots, \mathbf{x}_n\}$ as the *set* formed by $\mathbf{x}^n$, and let $E$ be the event on $\mathbf{x}^n$ such that $\mathbf{x}_0 \in A$ and $|A| \leq \frac{d}{2} + 1$. We now condition on the event $E$ occurring. This implies that the *out-of-sample* mass $I(q, \mathbf{x}^n) \geq \frac{\epsilon'}{2}$, since $|\mathcal{T}\backslash A| \geq \frac{d}{2}$ and $q$ is uniform when restricted to $\mathcal{T}\backslash\{\mathbf{x}_0\}$. By condition 2 of Definition 3 with $\gamma = 0$ [2], we have $I(\hat{p}_n, \mathbf{x}^n) = \sum_{\mathbf{x}\in\mathcal{X}\backslash A} \hat{p}_n[\mathbf{x}] \geq I(q, \mathbf{x}^n) \geq \epsilon'/2$. Note that the conditional distribution $\mu \mid \mathbf{x}^n$ restricted on $\mathcal{T}$ is exactly uniform over $\mathcal{C}' = \{\mathcal{T}' \in \mathcal{C} : A \subset \mathcal{T}'\}$. We have, for any $\mathbf{x} \notin A$ that

$$\mathbb{E}_{(q,\mathcal{T})\sim\mu|\mathbf{x}^n}[1\{\mathbf{x} \notin \mathcal{T}\}] = \mathbb{E}_{\mathcal{T}'\sim\mathcal{C}'}[1\{\mathbf{x} \notin \mathcal{T}'\}] = \frac{2d - |A|}{2d} \geq \frac{2d - d/2}{2d} = \frac{3}{4}.$$

This implies, for all $\mathbf{x}^n \in E$ that:

$$\mathbb{E}_{(q,\mathcal{T})\sim\mu|\mathbf{x}^n}[\mathsf{hall}(\hat{p}_n, \mathcal{T})] = \mathbb{E}_{(q,\mathcal{T})\sim\mu|\mathbf{x}^n}\left[\sum_{\mathbf{x}\in\mathcal{X}} \hat{p}_n[\mathbf{x}]1\{\mathbf{x} \notin \mathcal{T}\}\right]$$

$$\geq \mathbb{E}_{(q,\mathcal{T})\sim\mu|\mathbf{x}^n}\left[\sum_{\mathbf{x}\in\mathcal{X}\backslash A} \hat{p}_n[\mathbf{x}]1\{\mathbf{x} \notin \mathcal{T}\}\right]$$

$$\geq \frac{3}{4}\sum_{x\in\mathcal{X}\backslash A} \hat{p}_n[\mathbf{x}] \geq \frac{3\epsilon'}{8}.$$

We now observe that the distribution $q$ assigns only $\epsilon'$ probability mass on $\mathcal{X}\backslash\{\mathbf{x}_0\}$. The expected number of elements in $\mathbf{x}^n$ *not* equal to $\mathbf{x}_0$ is upper bounded by $\epsilon'n \leq \frac{d}{4}$. This implies, by Markov inequality, that w.p. $\geq \frac{1}{2}$ we have $|A| \leq \frac{d}{2}$ and $\mathbf{x}_0 \in A$, therefore $\Pr[E] \geq \frac{1}{2}$. Putting everything together, we have

$$\mathbb{E}_{(q,\mathcal{T})\sim\mu}\mathbb{E}_{\mathbf{x}^n\sim q}[\mathsf{hall}(\hat{p}_n, \mathcal{T})] = \mathbb{E}_{\mathbf{x}^n}\mathbb{E}_{(q,\mathcal{T})\sim\mu|\mathbf{x}^n}[\mathsf{hall}(\hat{p}_n, \mathcal{T})]$$

$$\geq \frac{1}{2}\mathbb{E}_{\mathbf{x}^n}[\mathbb{E}_{(q,\mathcal{T})\sim\mu|\mathbf{x}^n}[\mathsf{hall}(\hat{p}_n, \mathcal{T})] \mid E] \geq \frac{3\epsilon'}{16}.$$

Taking $\epsilon' > \frac{32}{3}\epsilon$, we have the *expected* hallucination rate $> 2\epsilon$. Since hallucination rate is $\leq 1$, we have w.p. $\geq \epsilon$, the hallucination rate is $> \epsilon$. Meaning that, for any sample size $n \leq \frac{d}{4\epsilon'} = \frac{3}{128}\frac{d}{\epsilon}$, *no* learner can achieve both conditions of Definition 3 for $\delta \leq \epsilon$. This completes the proof.

---

[2]We take $\gamma = 0$ for simplicity, however, our argument holds for general $\gamma > 0$ as well.

# B  ADDITIONAL LOWER BOUNDS

In this appendix, we provide a lower bound for improper non-hallucinating learning when the Shannon entropy is chosen as the information measure.

**Theorem 5.** *For any number $d \in \mathbb{N}$, there exists a concept class $\mathcal{C}$ of VC-dimension $d$ such that the sample complexity of $0$-approximately (improper) non-hallucinate learning $\mathcal{C}$ w.r.t. the Shannon entropy function $H$ is lower bounded by $\Omega(d)$ for all sufficiently small $\epsilon, \delta > 0$.*

*Proof.* We now take $\mathcal{X} = [d]$ and define $\mathcal{C}$ to be a collection of subsets of $\mathcal{X}$ such that $\forall \mathcal{T} \in \mathcal{C}$, $|\mathcal{T}| = d/2$ and

$$\forall \mathcal{T}_1 \neq \mathcal{T}_2 \in \mathcal{C}, \ |\mathcal{T}_1 \cap \mathcal{T}_2| \leq \frac{d}{4}.$$

It can be shown that such a collection exists and $|\mathcal{C}| \geq \sqrt{\frac{1}{4d}} e^{d/16}$, see e.g., (Wu et al., 2023, Thm. D.1) for a proof. Moreover, the VC-dimension of $\mathcal{C}$ is at most $d$. For any $\mathcal{T} \in \mathcal{C}$, we denote $q_{\mathcal{T}}$ as the uniform distribution over $\mathcal{T}$. Now, in order for the Definition 3 to hold, for any $(\mathcal{T}, q_{\mathcal{T}})$ the learner must produce a distribution $\hat{p}$ such that: (1) $\hat{p}[\mathcal{T}] \geq 1 - \epsilon$; and (2) the Shannon entropy $H(\hat{p}) \geq \log(d/2)$. We claim that for any other $\mathcal{T}' \in \mathcal{C}$ that differs from $\mathcal{T}$, one must have $\hat{p}[\mathcal{T}'] < 1 - \epsilon$, for any $\epsilon \leq 0.07$. To see this, we split the support of $\hat{p}$ into 3 parts $A_1 = \mathcal{T} \backslash \mathcal{T}'$, $A_2 = \mathcal{T} \cap \mathcal{T}'$ and $A_3 = [d] \backslash \mathcal{T}$. It is clear that $\hat{p}[A_3] \leq \epsilon$. Assume now that $\hat{p}[\mathcal{T}'] \geq 1 - \epsilon$, then one must have $\hat{p}[A_2] \geq 1 - 2\epsilon$. By expressing the entropy of $\hat{p}$ conditioning on the $A_1, A_2$ and $A_3$ we have

$$H(\hat{p}) = \sum_{i \in \{1,2,3\}} \hat{p}[A_i] \log \frac{1}{\hat{p}[A_i]} + \hat{p}[A_i] \cdot H(\hat{p} \mid A_i), \tag{10}$$

where $H(\hat{p} \mid A_i)$ is the entropy of the conditional distribution of $\hat{p}$ on $A_i$. Observe that $H(\hat{p} \mid A_1) \leq \log d/2$, $H(\hat{p} \mid A_2) \leq \log d/4$ and $H(\hat{p} \mid A_3) \leq \log d$, since uniform distribution maximizes entropy. Moreover, our previous discussion yields that $\hat{p}[A_1], \hat{p}[A_3] \leq \epsilon$ and $\hat{p}[A_2] \geq 1 - 2\epsilon$. Therefore, the RHS of (10) is upper bounded by (via Lagrangian multiplier method that the maximum attains on the boundary $\hat{p}[A_1] = \hat{p}[A_3] = \epsilon$ and $\hat{p}[A_2] = 1 - 2\epsilon$ whenever $\epsilon \leq 0.18$):

$$\text{RHS} \leq \underbrace{\epsilon \log d/2 + \epsilon \log \frac{1}{\epsilon}}_{\text{Contributed by } A_1} + \underbrace{(1 - 2\epsilon) \log d/4 + (1 - 2\epsilon) \log \frac{1}{(1 - 2\epsilon)}}_{\text{Contributed by } A_2} + \underbrace{\epsilon \log d + \epsilon \log \frac{1}{\epsilon}}_{\text{Contributed by } A_3}$$

$$= \log d + 2\epsilon \log \frac{1}{\epsilon} + (1 - 2\epsilon) \log \frac{1}{1 - 2\epsilon} + 3\epsilon - 2.$$

Sine $H(\hat{p}) \geq \log d - 1$, we have $-2\epsilon \log \epsilon - (1 - 2\epsilon) \log(1 - 2\epsilon) + 3\epsilon \geq 1$. Rewrite the expression, this implies $h(2\epsilon) + 5\epsilon \geq 1$, where $h(\cdot)$ is binary entropy function. Numerical computation yields that this can happen only when $\epsilon \geq 0.07$.

Now, the above discussion implies that any successful learner that satisfies Definition 3 with $\epsilon \leq 0.07$ must be able to identify the distribution $q_{\mathcal{T}}$ for all $\mathcal{T} \in \mathcal{C}$ by observing only on their $i.i.d.$ samples. We now derive a lower bound for such a identification problem using the Fano's inequality. Denote $\bar{q}$ as the uniform distribution over $[d]$, the Fano's inequality Polyanskiy & Wu (2022) implies that the error probability of the identification problem of sample size $n$ is lower bounded by

$$1 - \frac{1}{\log |\mathcal{C}|} \left( \sup_{\mathcal{T} \in \mathcal{C}} n \cdot \mathsf{KL}(q_{\mathcal{T}} || \bar{q}) + \log 2 \right) = \delta.$$

Direct computation yields that $\mathsf{KL}(q_{\mathcal{T}} || \bar{q}) \leq \log 2$ for all $\mathcal{T} \in \mathcal{C}$. Therefore, an $O(1)$ error probability lower bound holds as long as $n \ll \log |\mathcal{C}|$, i.e., for sufficiently small $\delta > 0$ we have the sample complexity $n \geq \Omega(\log |\mathcal{C}|) \geq \Omega(d)$ since $|\mathcal{C}| \geq \sqrt{\frac{1}{4d}} e^{d/16}$ as discussed above. □

