# OpenReview forum: "No Free Lunch: Fundamental Limits of Learning Non-Hallucinating Generative Models"
_ICLR.cc/2025/Conference — ICLR 2025 Poster_

### Official Review · Reviewer_NhVT · 2024-10-27

**Soundness:** 3
**Presentation:** 2
**Contribution:** 2
**Rating:** 5
**Confidence:** 2

**Summary:**

The paper develops a theoretical framework to explore the limitations of training generative models that do not hallucinate. It finds that learning non-hallucinating models based purely on truthful training data is statistically impossible without incorporating inductive biases. The study further introduces scenarios where non-hallucinating learning is achievable by using improper learners or by restricting the hypothesis space with finite VC-dimension concept classes.

**Strengths:**

1. The paper explores the inherent challenges and limitations of achieving non-hallucinating learning, introducing the theoretical impossibility of agnostic proper learning.
2. Through Theorems 1-4, the paper offers new theoretical insights, such as the impossibility of agnostic guarantees and the role of concept classes with finite VC-dimension in enabling non-hallucinating learning.
3. The paper provides upper and lower sample complexity bounds to balance model generalization and non-hallucination.

**Weaknesses:**

The framework proposed is primarily conceptual, and may need practical guidance for deploying non-hallucinating generative models in real-world applications.

**Questions:**

1. Your framework assumes that inductive biases can be introduced through concept classes. How is it applied to generative models in real-world scenarios?
2. The VC-dimension plays a key role in characterizing non-hallucinating learnability, but for complex models like transformers, estimating the VC-dimension is challenging. Are there alternative complexity measures or proxies you considered?
3. Can the framework be extended to evaluate hallucinations across multimodal generative models, such as vision-language models? If so, how would you do that?

---

> ### Author Response · Authors · 2024-11-19
>
> We thank the reviewer for acknowledging our contributions, as well as for providing the detailed review. We now address the main questions:
>
>
>
> 1. **Incorporating inductive biases in practice:** As we have shown in Theorem 2, non-hallucinating learning is possible by restricting the concept class to a class with finite VC-dimension. This demonstrates that if the "non-hallucination" set can be described by a neural network, then an improper learner can construct a model that avoids hallucinations, while generalizing to the information provided by the data distribution. This is analogous to approaches in RLHF, where the reward model can be viewed as a concept function that characterizes what is "hallucination." However, our approach differs in that our "reward model" is learned solely from the training data without human input. The precise learning rule can be extracted from (7), albeit computationally inefficient.
>
>
>
> 2. **Complexity measures:** We would like to clarify that having a finite VC-dimension is not a stringent requirement, since for any neural network with a binary outcome and any reasonable activation function (such as ReLU), the VC-dimension can be upper-bounded by the size of the network. However, it is quite possible that there might be tighter complexity measures that characterize the sample complexity (such as Rademacher complexity). Since our paper is an initiation of this broad study, we have elected to use the VC-dimension as it is more natural and well-understood. As we have pointed out in our paper, investigating other complexity measures particularly tailored to the specific information measure would be of significant interest for future research.
>
>
>
> 3. **Extending to other scenario:** The primary purpose of our paper is to provide a clear exposition of understanding the *fundamental challenges* in learning a non-hallucinating model. To this end, it is beneficial to start with the simplest and clearest model to get the intrinsic insight. However, we believe it would be interesting to investigate various extensions to other scenarios, including incorporating context and dealing with multimodal models.

---

### Official Review · Reviewer_m9LK · 2024-10-29

**Soundness:** 3
**Presentation:** 2
**Contribution:** 3
**Rating:** 8
**Confidence:** 2

**Summary:**

The paper studies learning hallucination-free models from a learnng theory perspective and provides multiple negative and positive results, the latter for cases when inductive biases are present.

**Strengths:**

**I am by far not an expert in learning theory, so my theoretical understanding of this paper is limited. Still, I can appreciate the authors' contributions:**

- the paper's topic is highly relevant, interesting, and the results (seem to be) strong
- the setup is clear from the introduction (though it's maybe too technical to be placed into the introduction)
- the technical results have nice intuitive explanations (even for someone who is not an expert in learning theory)

**Weaknesses:**

**My main point is that the presentation lacks clarity and detailed explanations of the steps taken. Note, however, that I am no expert in learning theory. Due to this fact, I choose a conservative score, but I am willing to reconsider during the rebuttal.**

### Major points
- several details are deemed "simple/clear/easy to verify" and left out, and non-standard abbreviations are not resolved (even though the reader might be able to guess them)
- the proofs could have been moved to the appendix, for the main text providing proof sketches would have been sufficient (and this would provide space to elaborate on intuition/notation) - by sketch I mean an intuitive description of the steps, not as technical as the one provided for Theorem 4.

### Minor points
- a Figure 1 for an intuitive overview would improve the paper
- L55: what is an instance space?
- the introduction is too long and already contains technical details (and readers probably won't expect to find these in the introduction), please consider restructuring the paper
- also, though the contributions list is helpful, it is too long to be considered as a "summary"
- L117: "Our results demonstrate"
- Eq (3): please specify what the subscript TV stands for.
- L160: what does 3-agnostic mean? Does the "3" relate to Eq (3) or something else?
- Fact 1:
	- please don't call this a "simple fact."
	- what is "w.p."?
- L195: what is "w.h.p."?
- Example 1: I appreciate that you provide an example, though skipping steps by writing "It is easy to verify" doesn't help the reader. Please elaborate.
- L317: what do you mean by "measured competitively?"

**Questions:**

- You use plausibility, validity, and factuality (and their negations) to describe hallucinations. Though for me the definition of hallucination rate was very clear, the definition in the abstract _("plausible but invalid")_ seems not to capture all aspects of hallucinations. I'd argue that implausible (and false) outputs also belong to hallucinations. Could you please clarify which one you mean?
- Out of curiosity, how does your intuition relate to impossibility results in the identifiability literature (eg,  http://proceedings.mlr.press/v97/locatello19a.html), which also state that some further assumption ("inductive bias") is needed for identifiability?

---

> ### Author Response · Authors · 2024-11-19
>
> We thank the reviewer for appreciating our contributions, as well as for providing the detailed review. We now address the main questions:
>
>
>
> 1. **Clarify abbreviations:** We have clarified the abbreviations in the revised paper and removed the "simple/clear/easy to verify" statements where appropriate.
>
>
>
> 1. **Clarity of Proof:** We thank the reviewer for the suggestion. We have moved the proof of Theorem 3 to the appendix and replaced it with a proof sketch. Additionally, we clarified the proof sketch of Theorem 4 with a more intuitive argument. Please refer to the parts highlighted in teal in the revised paper. We hope this clarifies the high-level proof ideas.
>
>
>
> 2. **Minor points:** We thank the reviewer for the suggestions; we have incorporated them in the revised version as appropriate. We clarify some of the key points below:
>
>    - The instance space in our context means all sentences (i.e., all finite sequences of tokens).
>
>    - For the introduction, we appreciate the reviewer's suggestion. We include the formal setup in the introduction to provide sufficient information for the reader to quickly understand the "real" problem we address in the paper, which is a common practice in theoretical papers.
>
>    - TV stands for Total Variation distance.
>
>    - "3-agnostic" means that $\alpha = 3$ in Eq (3).
>
>    - "w.p." stands for "with probability."
>
>    - "w.h.p." stands for "with high probability."
>
>    - In Example 1, statement (1) follows from the fact that $q$ is supported solely on $A$ and $A \subset \mathcal{T}$; (2) follows from the adversarial selection of $\mathcal{T}$; and (3) follows because the distribution $p_i$, which differs from $\hat{p}_n$, must have a hallucination rate of 0.
>
>    - "Measured competitively" means that we do not quantify the hallucination rate absolutely for the learned model but compare it to the best model in a hypothesis class.
>
>
>
> **Questions:**
>
>
>
> 1. We apologize for the confusion. In our formulation, the "facts set" is what should be considered the "non-hallucination set," which consists of all sentences excluding both "plausible but wrong" and "clearly wrong" sentences. Please note that our result does not rely on the exact definition of what "hallucination" means semantically; instead, our result aims to characterize the learnability of relating the "hallucination" in the learned model to that in the training data. We have revised the abstract from "plausible but invalid" to "not grounded in the underlying facts".
>
>
>
> 2. **Comparison to Locatello et al. (2019):** Indeed, the requirement of "inductive bias" is common in many machine learning paradigms, such as the disentangled representation learning in Locatello et al. (2019). What is unexpected (perhaps surprisingly) in our result is that we demonstrate  that non-hallucinating learning is "strongly impossible" even when measured competitively (i.e., by comparing to the best model in a hypothesis class), which contrasts with classical PAC learning.

---

> > ### Comment · Reviewer_m9LK · 2024-11-23
> >
> > Apologies for my late response and thank you for your effort to address my concerns! I have checked your revised submission and am generally pleased with your changes. Based on this and the other reviews (and your corresponding responses).
> >
> > Particularly, I concur with the summary of Reviewer S3iM in the [note](https://openreview.net/forum?id=OwNoTs2r8e&noteId=UTz2gNWuSR).
> > I would only add one point regarding **Figure 1**, but nevertheless, I **raised my score**: first, I greatly appreciate you adding this to the revision. Second, please provide a detailed caption (to make it self-contained) describing all the notations involved (people often first look at Figure 1 only to decide whether they want to read the paper). In case you have troubles with space, you could use a `wrapfigure` environment, as the figure design is such that currently there is a lot of empty space.

---

> > > ### Author Response · Authors · 2024-11-23
> > >
> > > We are grateful to the reviewer for the constructive comments and for increasing the score. In the revised manuscript, we have included a detailed caption for the figure.

---

> > > > ### Comment · Reviewer_m9LK · 2024-11-23
> > > >
> > > > The caption is very nice, thank you! All in all, you did a great job!

---

### Official Review · Reviewer_S3iM · 2024-11-02

**Soundness:** 3
**Presentation:** 2
**Contribution:** 3
**Rating:** 8
**Confidence:** 3

**Summary:**

This paper considers the problem of hallucinations in generative models from a learning theory lense. The authors highlight that avoiding hallucinations poses a fundamental problem: On the one hand, a model should generalize beyond its training distribution, but on the other hand, undesired implausible artefacts (hallucinations) have to be avoided without a strict characterization of what is implausible. The authors formalize this issue and are able to show that without inductive biases that restrict the set of plausible samples, it is impossible to avoid hallucinations. This is an important result that will be of interest to the wider community, especially also for practicioners who attempt to avoid hallucinations via empirical means. The authors go on to explore how inductive biases can facilitate learning without hallucinations, but highlight that this problem remains challenging.

**Strengths:**

The paper, to the best of my knowledge, introduces a novel take on hallucinations in generative models and is the first to show such a strong impossibility result. While the paper remains purely theoretical and doesn't offer an implementation for cases in which hallucination-free learning is in principle possible (as the authors also admit), I believe this result will nontheless be informative for the community. Theorem 4 and §4.2 in general make a first step in understanding when avoiding hallucinations is possible.

The paper is thorough in setting up the problem and does a good job of elucidating the importance of its results.

**Weaknesses:**

The paper is quite dense and might be inaccessible to a larger audience, especially to practitioners for whom this result might be very relevant. I would highly suggest the authors to add, e.g., a short paragraph after each theorem that translates the abstract results into specific examples and gives a higher-level intuition. E.g. consider a simple sample task using real data: Which assumptions of the theorem will be met? Are there some implications the theorem makes that would be violated but can be considered to be not very impactful in practice? I believe some of the proofs could be moved to an appendix to make space for this.

**Questions:**

# Questions
- LL178: Is point (ii) even always possible in principle, i.e., will the learned model always generalize given sufficient training samples?

# Minor suggestions
- LL65: What is $\mathcal X^*$?
- LL130: typo "being" → "been"
- LL151: while it is technically correct to call $\mathrm x^n$ a "sample", this might confuse readers to assume $\mathrm x^n$ is a single point rather than a set of points. I recommend using a slightly different terminology
- Eq. 3: $\| \cdot \|_\text{TV}$ what does this notation mean? I assume based on the paragraph below it is "total variation distance"?
- §1, §2: as far as I can see, the term PAC learning is never introduced
- LL167: What does "w.p." stand for?
- LL195: What does "w.h.p." stand for?
- LL336: Should be "note that, here, ..."

Generally, I recommend double-checking that abbreviations and notations used are introduced whenever they first appear. Where not absolutely necessary, avoiding abbreviations increases readability and makes the paper more accessible to readers from other subfields.

---

> ### Author Response · Authors · 2024-11-19
>
> We thank the reviewer for appreciating the significance and relevance of our work, as well as for providing the detailed review. We now address the main questions:
>
>
>
> 1. **Clarity of Writing:** We thank the reviewer for the suggestions. We have added additional discussions as recommended and clarified some of the technical proofs in our revised version (highlighted in teal). Specifically, we included a discussion after Theorem 2 on the relationship of our approach with RLHF and added an explanation after Theorem 4 on how to verify the conditions stated in Definition 4 empirically. We hope this clarifies how our theoretical results can be interpreted in practice.
>
>
>
> **Questions:**
>
>
>
> 1. **The impossibility of point (ii) in LL178**: Indeed, this point precisely highlights the challenge of requiring the model to generalize to the demonstration distribution under total variation. This is precisely the reason that motivated us to define generalizability via the "information measure" in Definition 4, rather than through PAC learning under total variation. Please note that in Definition 4, the learned distribution is not necessarily close to the demonstration distribution under total variation.
>
>
>
> 2. **Minor suggestion:** We are grateful to the reviewer for catching the typos; they have been fixed in the revised version.

---

> > ### Comment · Reviewer_S3iM · 2024-11-20
> >
> > I appreciate the author's effort in improving the clarity of writing and answering my questions. Since the paper's contributions are clear to me on a high level and seem interesting and relevant to the community, **I recommend acceptance** and have adjusted my score accordingly.
> >
> > That said, I still think the manuscript is not _great_ in terms of structure, clarity, and accessibility. I can see that other reviewers raised the same concerns, and the authors have addressed some but not all of those issues. In particular:
> >
> > - I concur with reviewer m9LK that the introduction is too long and technical, and would recommend moving the technical setup and definitions to §2 while keeping a high-level overview and outline of the paper (especially how the theorems build upon each other) in §1.
> > - I second the recommendation from reviewer m9LK that an introductory overview figure would improve accessibility of the paper.
> > - In my opinion, the paper should be self-contained even for a reader without expertise in learning theory. In this sense, PAC learning could be given a bit more room in §2 (it's still unclear what PAC stands for, and the authors later draw on a lot of intuition about this learning paradigm that is not necessarily clear to the reader).
> > - I still recommend to also replace the proof for Thrm. 1 with a proof sketch and instead use the space to give the reader a better intuition of the impact of this theorem and walk through some examples.
> > - Other reviewers also seemed confused by the frequent abbreviations. Where not absolutely necessary, I recommend avoiding abbreviations to make the text easier to parse.
> >
> > While these issues should not prevent the paper from being accepted, I urge the authors to consider them. It is in the authors' best interest to make the paper as accessible as possible to readers to ensure that the community can and will build on this result.
> >
> > Lastly, I want to highlight that I disagree with the assessment of the work by reviewer fY7z as "extremely flawed". While I would have welcomed experiments on a toy setting to elucidate the claims, I concur with the authors that theoretical results are contributions in their own right.

---

> > > ### Author Response · Authors · 2024-11-20
> > >
> > > We thank the reviewer for appreciating our contributions and for recommending acceptance. We have included a figure in the revised manuscript to illustrate the overall idea of our non-hallucinating learning paradigm and to clarify the abbreviation of the term "PAC learning". This certainly improves our presentation, and we appreciate your constructive comments.

---

### Official Review · Reviewer_mw5f · 2024-11-02

**Soundness:** 3
**Presentation:** 2
**Contribution:** 3
**Rating:** 8
**Confidence:** 2

**Summary:**

The authors examine the problem hallucinations in generative models from a learning theory perspective. They provide a formal setup for the problem, define the hallucination rate formally (as the probability mass assigned to samples not in some true fact subset) and characterize the importance of distinguishing between proper/improper learning for this paradigm. They then provide three main theoretical results:
- Proper learning (i.e. restricted to a hypothesis class) without hallucination is impossible.
- Non-hallucinating learning that generalizes is possible given an improper learner and restrictions on the VC-dimension of the fact set $\mathcal{T}$. They provide sample complexity bounds for this case.
- Proper learning with a VC concept class is possible as long as $q$ (the data generating process) is sufficiently informative.

**Strengths:**

- The work provides a novel and important contribution that addresses a particularly relevant problem (namely that of hallucinations in generative models) which has not been thoroughly studied from a theoretical perspective.
- The paper is generally well-written and structured.
- The construction of the counter-example for Theorem 1 is interesting and clear.
- Section 4.1 is particularly valuable studying the case of a model that can generalize without hallucinating.

**Weaknesses:**

- The work could benefit for a more in-depth comparison with Kalai & Vempala (2024).
- The impossibility results in section 3 seem to rely on the pathological case of an uninformative $q$. It would be interesting to investigate the case where $q$ is sufficiently informative as per definition 4.
- As acknowledged by the authors, the work is mostly conceptual and a first step in this research direction.
- The messaging on the key takeaways from the paper could be improved (especially for practitioners not particularly interested in learning theory).

Nitpicks:
- The work could do with less italicized words (feels like there's at least 1-2 every sentence).
- Some typos such as:
  - Line 245: For the -> the
  - Line 294: Complited -> Completed
  - Line 326: Leaner -> Learner

**Questions:**

- Theorem 1 is written as specifically for the case of a hypothesis class of 2, do you think the result would still hold for hypothesis classes of arbitrary sizes?
- I'm a bit confused by how exactly Example 3 differs from Example 1?

---

> ### Author Response · Authors · 2024-11-19
>
> We thank the reviewer for appreciating the novelty and significance of our work, as well as for providing the detailed review. We now address the main questions:
>
>
>
> 1. **Comparison with Kalai & Vempala (2024):** The negative result of Kalai & Vempala (2024) is mostly comparable to our Theorem 3, which shows a lower bound on the hallucination rate based on the notion of "calibration" of the learned distribution. Here, "calibration" can be viewed as a different way of quantifying the "generalizability" of the learned distribution. In our work, we define generalizability via the concept of information measure maximization. Technically, however, our lower bound is not directly comparable to theirs.
>
>
>
> 2. **Sufficient informativeness of $q$ in Section 3:** Indeed, any given instance $q$ constructed in Theorem 1 is sufficiently informative if we specify the concept class as all the possible $\mathcal{T}$ constructed therein. To see this, we observe that if $q \sim \mu_i$, then $p_i\in \mathcal{P}$ always satisfies the condition in Definition 4 for $\xi < \frac{1}{2}$. The reason why Theorem 1 does not contradict Theorem 4 is that the concept class used in Theorem 1 has an *infinite* VC-dimension.
>
>
>
> 3. **Key takeaways:** The main message we would like to convey is that there can be inherent limitations in learning a non-hallucinating generative model, even when the entire training set is *non-hallucinating*. This limitation is inherent in the sense that it is independent of the exact form of how one defines "hallucination". Therefore, one should not expect to find a "universal" solution and should instead focus on integrating *factual* information itself into the learning process.
>
>
>
> **Questions:**
>
>
>
> 1. Indeed, for larger classes, the impossibility result may not hold. For instance, if we take $\mathcal{P}$ to be the class of all distributions, we reduce to the improper learning case. However, it is presently unclear to us how the transition occurs with respect to the size of the class.
>
>
>
> 2. Technically, Example 1 and Example 3 are nearly identical, with the only difference being the emphasis on the concept class size.
>
>
>
> 3. We thank the reviewer for identifying the typos; they have been corrected in the revised version.

---

> > ### Comment · Reviewer_mw5f · 2024-11-25
> >
> > I thank the authors for their clarifications, the contribution is clearer and my concerns have been largely addressed. I have raised my score accordingly (though with the caveat that I am not an expert on learning theory).

---

### Official Review · Reviewer_fY7z · 2024-11-04

**Soundness:** 1
**Presentation:** 2
**Contribution:** 1
**Rating:** 1
**Confidence:** 4

**Summary:**

-This paper is rather strange, because it seems to be about hallucination as a specific problem, but the theoretical claims just treat hallucination as a subset of examples.  It seems like the claims are about a more general phenomena of trying to learn examples which belong to a set.  In this case, set is characterized as a set of factual claims, but this property isn’t used.  In the technical results, it’s just a set.

Notes from reading the paper:
 -Hallucinations are outputs which are plausible but invalid.
  -Impossible to stop hallucinations using data, but need to leverage inductive biases about facts.
  -Paper is purely conceptual, but if claim is valid, seems very striking.
  -Let X be the set of sentences, while T is the set of facts, or sentences that describe true statements that are relevant.  The hallucination rate is the rate hall(p, T) of sentences generated which aren’t facts.
  -A demonstrator q is faithful wrt T if hall(q,T)=0.

**Strengths:**

The issue of conceptualizing and theorizing about hallucinations seems like an important and useful problem to address.

**Weaknesses:**

The paper seems extremely flawed.  First, it is a purely theoretical paper, with no analysis or experiments, even on toy models.  I think this is dubious, especially when the theoretical framework is speculative and not well established.  For example, even a small illustration on a real toy dataset would help to clarify the ideas of the paper.  Second, it doesn't seem like the analysis actually uses the nature of hallucinations in proving the result, which leads me to think that the paper could be written with a more general claim.

**Questions:**

Couldn’t a statement simply be non-factual, but not a hallucination.  Context also seems important. For example, a claim may be factual in a story but not in general.  Additionally, a question seems like a sentence which is non-factual but not a hallucination.

---

> ### Author Response · Authors · 2024-11-19
>
> We respectfully disagree with the reviewer's assessment of our work. The reviewer seems to have two main concerns: (i) the theoretical nature of our contribution, and (ii) that our formulation does not fully capture the semantic meaning of "hallucination." Regarding the first concern, our theoretical treatment, is, in fact, a key strength of our paper – we establish fundamental limitations on hallucination that are independent of empirical design choices and implementations. Concerning the second point, we would like to clarify that our formulation of the "facts set" is precisely what should be considered a "non-hallucination set," which is a general concept regardless of one's definition of "hallucination." Our work does not aim to characterize the *semantic* meaning of "hallucination"; rather, it focuses on establishing the *inherent statistical limits* of learning a non-hallucinating model that relies solely on the training sample, without external definitions of what constitutes "hallucination."
>
>
>
> To set the stage for our discussion, we reiterate our main contributions as follows:
>
>
>
> Our results reveal several inherent statistical challenges of learning non-hallucinating models, such as Example 1 and Theorem 1, which demonstrate that even in a class with only two models, a *universal* learner cannot learn a non-hallucinating model even if one exists in the class. This is a significant conceptual contribution (as acknowledged by reviewers S3iM and mw5f), showing that a non-hallucinating model cannot be learned "agnostically", in contrast to the standard PAC model. Additionally, we provide several approaches for incorporating "factual" information into the learning process to enable non-hallucinating learning. For example, Theorem 2 shows that if the "non-hallucination" set can be specified by a neural network, then an improper learner can learn a model that avoids hallucinations while generalizing to the information provided by the data distribution. This is analogous to approaches in RLHF, where the reward model can be viewed as our "concept function" that characterizes what is "hallucination." However, our approach differs in that our "reward model" is learned solely from the training data without human input. This is a significant contribution, as it puts forth a systematic approach with rigorously provable guarantees for non-hallucinating learning that relies solely on training data yet generalizes.
>
>
>
> **We now address the specific points raised:**
>
>
>
> **Q1: The paper seems extremely flawed. First, it is a purely theoretical paper, with no analysis or experiments, even on toy models.**
>
>
>
> **A1:** We strongly disagree with this claim. As acknowledged by all other reviewers (mw5f, S3iM, m9LK, NhVT), our paper presents novel, significant, and insightful theoretical results that advance the understanding of non-hallucinating learning within a rigorous theoretical framework. Our findings are derived through rigorous theoretical analysis and proofs, and we find the assertion that our paper is "extremely flawed" to be inappropriate. Historically, ICLR has accepted a large number of theoretical contributions – this is not an acceptable reason for rejecting a paper.
>
>
>
> **Q2: I think this is dubious, especially when the theoretical framework is speculative and not well established. For example, even a small illustration on a real toy dataset would help to clarify the ideas of the paper.**
>
>
>
> **A2:** As stated in our paper, our primary contribution is conceptual, aiming to understand the fundamental limits of non-hallucinating learning in a *principled* manner (i.e., invariant to specific experimental setups). Our work provides a comprehensive theoretical investigation, which we believe will be valuable to the ICLR community. Our theoretical analysis is completely sound (i.e., with rigorous proof) and novel (i.e., previously unexplored). This is a core strength of our paper.
>
>
> **Q3: Second, it doesn't seem like the analysis actually uses the nature of hallucinations in proving the result, which leads me to think that the paper could be written with a more general claim.**
>
>
> **A3:** Indeed, our analysis does not rely on any specific semantic definition of "hallucination"; this is the **key novelty** of our paper, as it enables us to analyze non-hallucinating learnability "relatively" that relies solely on the training data. However, our results do depend on the definitions of a "non-hallucinating model." We provide two characterizations using the concept of "hallucination rate" (which is agnostic to the semantic meaning of "hallucination"), specifically, the absolute and relative hallucination rates. Our proofs rely heavily on these definitions of "hallucination rate."

---

> > ### Author Response · Authors · 2024-11-19
> >
> > **Q4: Couldn’t a statement simply be non-factual, but not a hallucination? Context also seems important. For example, a claim may be factual in a story but not in general. Additionally, a question seems like a sentence which is non-factual but not a hallucination.**
> >
> > **A4:** We clarify that "hallucination" does not necessarily equate to the semantic notion of facts (for example, there may be non-factual sentences that are easily recognizable and thus not considered hallucinations). However, our results are independent of any specific definition of "hallucination." In contextual cases, the "facts set" could also be context-dependent (e.g., based on a prompt). For simplicity and clarity, in our theoretical exposition, we have not included contextual elements, similar to the approach in Kalai & Vempala (2024).

---

### Meta-Review · Area_Chair_Ea42 · 2024-12-04

**Metareview:**

This paper presents an impossibility result for non-hallucinating generative models and conceptually explains the role of inductive bias. Overall, while one reviewer was negative, the others stood up for the paper in the discussion, and I agree with their assessment. The direction is timely, and the definition and study of hallucinations are likely to inspire new work in learning theory and, hopefully, practice. The main criticisms addressed during the rebuttal concerned the presentation.

While I side with the majority of reviewers, I also want to say that I partially also agree with Reviewer fY7z. This paper could have had some experiments supporting the theoretical claims and showcasing how these results may lead toward a principled approach to address hallucinations, which would have made the contribution much stronger, in my opinion.

**Additional Comments On Reviewer Discussion:**

During the discussion period, mostly questions on the presentation of the paper and the clarity of the writing were discussed (and addressed by the authors). The strongly negative review concerned the complete lack of experiments, which I agree is a severe weakness, but in this case, I would say not a fatal flaw, as the theoretical formulation is standalone interesting. It is not fully clear to me, however, whether the theory is novel in itself or it is a standard application of PAC learning rephrased for this new application.

---

### Decision · Program_Chairs · 2025-01-22

Accept (Poster)